# ChatGPT versus human in generating medical graduate exam multiple choice questions—A multinational prospective study (Hong Kong S. A.R., Singapore, Ireland, and the United Kingdom)

**Billy Ho Hung Cheung**[1], **Gary Kui Kai Lau**[1], **Gordon Tin Chun Wong**[1], **Elaine Yuen Phin Lee**[1], **Dhananjay Kulkarni**[2], **Choon Sheong Seow**[3], **Ruby Wong**[4], **Michael Tiong-Hong Co**[1]*

1 L.K.S. Faculty of Medicine, University of Hong Kong, Hong Kong, Hong Kong S.A.R, 2 Department of Surgery, University of Edinburgh, Edinburgh, United Kingdom, 3 Department of Surgery, National University Cancer Institute Singapore, Singapore, Singapore, 4 Department of Surgery, University of Galway, Galway, Ireland

* mcth@hku.hk

## Abstract

### Introduction

Large language models, in particular ChatGPT, have showcased remarkable language processing capabilities. Given the substantial workload of university medical staff, this study aims to assess the quality of multiple-choice questions (MCQs) produced by ChatGPT for use in graduate medical examinations, compared to questions written by university professoriate staffs based on standard medical textbooks.

### Methods

50 MCQs were generated by ChatGPT with reference to two standard undergraduate medical textbooks (Harrison's, and Bailey & Love's). Another 50 MCQs were drafted by two university professoriate staff using the same medical textbooks. All 100 MCQ were individually numbered, randomized and sent to five independent international assessors for MCQ quality assessment using a standardized assessment score on five assessment domains, namely, appropriateness of the question, clarity and specificity, relevance, discriminative power of alternatives, and suitability for medical graduate examination.

### Results

The total time required for ChatGPT to create the 50 questions was 20 minutes 25 seconds, while it took two human examiners a total of 211 minutes 33 seconds to draft the 50 questions. When a comparison of the mean score was made between the questions constructed by A.I. with those drafted by humans, only in the relevance domain that the A.I. was inferior to humans (A.I.: 7.56 +/- 0.94 vs human: 7.88 +/- 0.52; p = 0.04). There was no significant

**Data Availability Statement:** All relevant data are within the paper and its Supporting information files.

**Funding:** No.

**Competing interests:** No.

difference in question quality between questions drafted by A.I. versus humans, in the total assessment score as well as in other domains. Questions generated by A.I. yielded a wider range of scores, while those created by humans were consistent and within a narrower range.

## Conclusion

ChatGPT has the potential to generate comparable-quality MCQs for medical graduate examinations within a significantly shorter time.

## Introduction

The workload of university medical staff is a pressing issue that requires attention [1], Medical staff are often tasked with multiple responsibilities that can include, but are not limited to, patient care, teaching, student assessment, research, and administrative work [2]. This heavy workload can be especially challenging for medical academic staff, who are also required to uphold the standard of undergraduate medical exams. The demand for exam quality has increased in recent years, as students and other stakeholders expect assessments to be fair, accurate, unbiased and aligned with the predefined learning objectives [3, 4].

In this context, the development of artificial intelligence (A.I.), machine learning, and language models offers a promising solution. A.I. is a rapidly developing field that has the potential to transform many industries, including education [5]. A.I. is a broad field that encompasses many different technologies, such as machine learning, natural language processing, and computer vision [6]. Machine learning is a subset of A.I. that involves training algorithms to make predictions or decisions based on the database [7]. This technology has been used in a variety of applications, including speech recognition, image classification, and language generation.

A large language model is a type of A.I. that has been trained on a massive amount of data and can generate human-like text with impressive accuracy [8]. These models are called "large" because they have a huge number of parameters, which are the weights in the model's mathematical equations that determine its behaviour. The more parameters a model has, the more information it can store and the more complex tasks it can perform.

ChatGPT is a specific type of large language model developed by OpenAI [9]. It is a state-of-the-art model that has been trained on a diverse range of texts, including news articles, books, and websites. This makes it highly versatile and capable of generating text on a wide range of topics with remarkable coherence and consistency. It is a pre-trained language model with knowledge up to 2021. However, it is possible to feed in relevant reference text by the operator, such that updated text or desired text outputs can be generated based on the operator's command and preference.

The potential implications and possibilities of using ChatGPT for assessment in medical education are significant. A recent publication confirmed that the knowledge provided by ChatGPT is adequate to pass the United States Medical Licensing Exam [10]. It is also believed that ChatGPT could be used to generate high-quality exam questions, provide personalized feedback to students, and automate the grading process, hence reducing the workload of medical staff and improving the quality of assessments [11]. This technology has the potential to be a game-changer in the field of medical education, providing new and innovative ways to assess student learning and evaluate exam quality.

Meanwhile, multiple-choice questions (MCQs) have been used as a form of knowledge-based assessment since the early twentieth century [12]. MCQ is an important and integral component in both undergraduate and post-graduate exams, due to its standardization, equitability, objectiveness, cost-effectiveness, and reliability [13]. When compared to essay-type of questions or short answer questions, MCQs also allow assessment of a broader range of content—each exam paper can include large numbers of MCQs [14]. This makes the MCQ format particularly suitable for summative final examinations. The major drawback to the MCQ format, however, is that high-quality questions are difficult and time-consuming to draft.

We hypothesize that advanced large language models, such as ChatGPT, can reliably generate high-quality MCQs that are comparable to those of an experienced exam writer. Here, with an updated large language model A.I. available, we aim to evaluate the quality of exam questions generated by ChatGPT versus those drafted by university professoriate staff based on international gold-standard medical reference textbooks.

## Methods

This is a prospective study to compare the quality of MCQs generated by ChatGPT versus those drafted by experienced university professoriate staff for medical exams (Fig 1). The study was conducted in February 2023. To allow a fair comparison, certain criteria are set for question developments.

1. The questions were designed to meet the standard for a medical graduate exam.

2. Only four choices were allowed for each question.

3. The questions were limited to knowledge-based questions only.

4. The questions were text-based only.

5. Topics regarding the exam context were set by an independent researcher before the design of the questions.

6. Both the professorial staff who designed the questions and the research operating ChatGPT were not allowed to view questions from the other side.

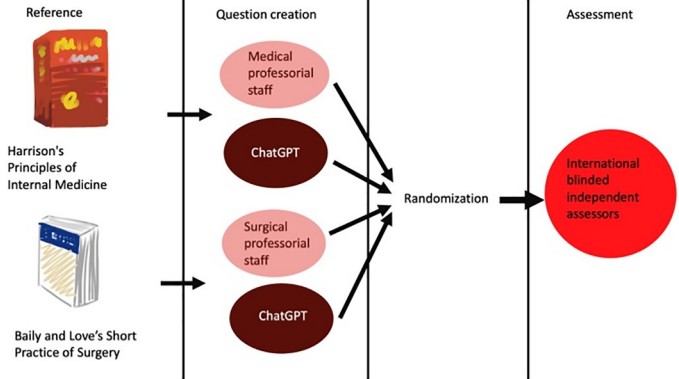

**Fig 1. Schematic diagram of the study design.**

7. Standardized, internationally-used textbooks, with Harrison's Principles of Internal Medicine 21th edition for medicine [15], and Baily and Love's Short Practice of Surgery 27th Edition for surgery [16], were used as the reference for question generation.

8. Distractors were allowed to refer from sources other than the original reference provided.

9. No explanation was required for the question.

ChatGPT plus [17] was used for question creation. This version is a small update to the standard ChatGPT version on the 1st of February 2023. According to the official webpage, ChatGPT Plus carries the ability to answer follow-up questions, and challenge incorrect assumptions. This also provides better access during peak hours, faster response time and priority to updates.

## Question construction

Fifty multiple choice questions (MCQs) with four options covering Internal Medicine and Surgery were generated by the ChatGPT with reference articles selected from the two reference medical textbooks.

ChatGPT was assessed in Hong Kong S.A.R. via a virtual personal network to overcome its geographical limitation [18]. The commander using the ChatGPT will only copy and paste the instruction "Can you write a multiple-choice question based on the following criteria, with the reference I am providing for you and your medical knowledge?" (also known as the prompt), the criteria and the reference text (selected from the reference textbook and input into ChatGPT interface as pure text input), to yield a question (S1 File). After sending the request to ChatGPT, a response will be automatically generated. Interaction is only allowed for clarification but not any modification. Questions provided by ChatGPT will be directly copied and used as the output for assessment by the independent quality assessment team. Moreover, a new chat session was started in ChatGPT for each entry to reduce memory retention bias.

The duration of work by A.I. was defined by the response time that ChatGPT needed to generate the questions once the prompt and the material were given, and the time for the operator to copy and paste the questions was not included. The duration of work by ChatGPT and humans was documented and compared.

Another 50 MCQs with four options were drafted by two experienced university professoriate staff (with more than 15 years of clinical practice and academic experience in Internal Medicine and Surgery, respectively) based on the same textbook reference materials.

They were given the exact instructions as the criteria mentioned above. Additionally, while they were allowed to refer back to additional textbooks or online resources, they were not allowed to use any A.I. tool to assist their question writing.

## Blinded assessment by a multinational expert panel

All 100 MCQs were individually numbered and randomized using a computer-generated sequence.

A multiple-choice item consists of the stem, the options, and any auxiliary information [12]. The stem contains context, content, and/or the question the student is required to answer. The options include a set of alternative answers with one correct option and one or more incorrect options or distractors. Auxiliary information includes any additional content, in either the stem or option, required to generate an item. The incorrect options to an MCQ are known as distractors; they serve the purpose of diverting non-competent candidates away

from the correct answer, which they serve as an important hallmark of a high-quality question [19].

The question set was then sent to five independent international blinded assessors (From the United Kingdom, Ireland, Singapore and Hong Kong S.A.R.) for assessment of the question quality (who were also the authors of this study as well). Members of the international panel of assessors are experienced clinicians with heavy involvement in medical education in their locality.

An assessment tool with five domains is specifically designed after literature review of relevant metrics for MCQ quality assessment in this study [20–23]. Specific instructions were given to the assessors for the meaning of each domain. These include (I) Appropriateness of the question, defined as if the question is correct, appropriately constructed with appropriate length and well-formed; (II) Clarity and specificity, defined as if the question is clear and specific without ambiguity, its answerability and without being under- or over-informative; (III) Relevance, defined as the relevance to clinical context; (IV) Quality of the alternatives & discriminative power for the assessment of the alternative choices provided; and (V) Suitability for graduate medical school exam focused on the level of challenge including if the question higher order of learning outcomes such as application, analysis and evaluation, as elaborated by Bloom and his collaborators [24]. The quality of MCQs was objectively assessed by a numeric scale of 0–10. "0" is defined as extremely poor, while "10" means that it is at the quality of a gold-standard question. The total score of this section ranges from 0 to 50.

In addition, the assessors were also asked to determine if the questions were constructed by A.I. or by humans, in which they were blinded about the total number of questions created by each arm. In addition, G.P.T. -2 Output Detector, an A.I. output detector [25] was also used to predict as if the question was written by A.I. or a human. This detector assigns a score between 0.02 and 99.98% to each question, with a higher score indicating a greater likelihood that the question was constructed by A.I.

## Statistical analysis

All data were prospectively collected by a research assistant and computerized into a database. All statistical analyses were performed with the Statistical Product and Service Solution (SPSS) version 29. A comparison was made between questions created by A.I. and by humans. Student T test or Mann-Whitney U test was used for the comparison of continuous variables for the five domains individually and combined. P value of less than 0.05 were considered statistically significant. Chi-squared test or Fisher's exact test were used to compare discrete variables, namely the perception of the assessor whether a question was produced by A.I. or human.

A paired t-test was performed to assess for systematic differences between the mean measures of each rater (including G.P.T. -2 Output Detector).

## Results

### Question construction

The question writing was performed by ChatGPT on two separate dates, 11th and 17th February. The work was carried out with stable internet via Wifi at a minimum of 15.40 Mbps for downloads and 11.96 Mbps for update speed.

The total time required for ChatGPT to create the 50 questions was 20 minutes 25 seconds, while it took two human examiners a total of 211 minutes 33 seconds (84 minutes 56 seconds for the surgical examiner and 126 minutes 37 seconds for the medical examiner).

**Table 1. Mean score of each domain.**

|  | Mean (± SD) | Range |
|---|---|---|
| Appropriateness of the question | 7.78 ± 0.74 | 5.40–9.80 |
| Clarity and specificity | 7.63 ± 0.69 | 5.60–9.20 |
| Relevance | 7.72 ± 0.77 | 5.60–9.20 |
| Quality of the alternatives & discriminative power | 7.31 ± 0.65 | 5.60–7.31 |
| Suitability for graduate medical school exam | 7.32 ± 0.84 | 5.00–9.20 |
| Total score | 37.76 ± 3.34 | 27.20–46.20 |

## Assessment of question quality

The results of the assessment by the independent blinded assessors were summarized in Table 1. We can see that the overall score was satisfactory with a mean score of each domain above 7.

When a comparison of the mean score was made between the questions constructed by A.I. with those constructed by humans, only in the relevance category that the A.I. was inferior to humans (A.I.: 7.56 ± 0.94 vs human: 7.88 ± 0.52; p = 0.04, Table 2). There was no significant difference in other domains of question quality assessment. The same applies to the total scores between A.I. and humans (Table 1).

Questions generated by A.I. yielded a wider range of scores, while those created by humans were consistent and within a narrower range (Fig 2). A similar distribution was also observed across all five domains (Fig 3).

## A.I. vs human

The average scores of the questions based on the same reading material were compared between that generated by A.I. vs by human writers. A "win" denotes a higher average score. Questions generated by humans generally received a higher mark when compared to the A.I. counterpart (Table 3). However, AI-drafted questions outperformed human ones in 36% to 44% of cases across five assessment domains, including the total score.

In addition, when the questions generated by ChatGPT were reviewed, we observed few negative features, including minimal use of negative stem (only 14% (7/50), compared to 12% (6/50) by human examiners) with a lack of "except", "All/none of the above".

## Blinded guess of the question writer by a panel of assessors

Assessors were asked to deduce if the question was written by A.I. or humans, and the results are shown in Table 4. The results of the assessment by G.P.T. -2 Output Detector were also

**Table 2. Comparison of mean scores between questions generated by AI and human.**

|  | AI (± SD) | Human (± SD) | P |
|---|---|---|---|
| Appropriateness of the question | 7.72 ± 0.83 | 7.84 ± 0.65 | 0.45 |
| Clarity and specificity | 7.56 ± 0.81 | 7.69 ± 0.55 | 0.34 |
| Relevance | 7.56 ± 0.94 | 7.88 ± 0.52 | 0.04 |
| Quality of the alternatives & discriminative power | 7.26 ± 0.68 | 7.36 ± 0.61 | 0.46 |
| Suitability for graduate medical school exam | 7.25 ± 0.94 | 7.40 ± 0.72 | 0.39 |
| Total score | 37.36 ± 3.92 | 38.16 ± 2.62 | 0.23 |

SD = standard deviation

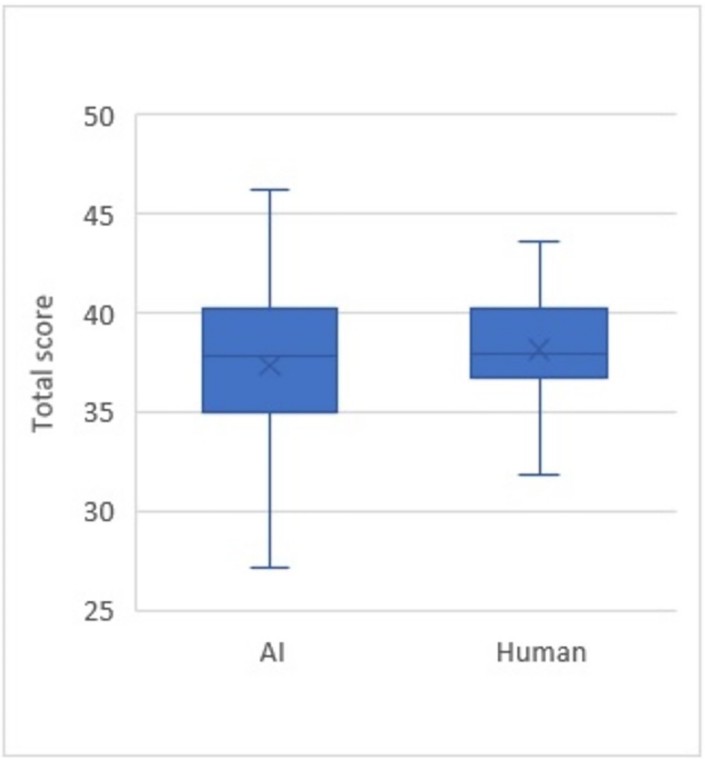

**Fig 2. Assessment scores of MCQ quality between A.I. and human.**

shown. The percentage of correct guesses was consistently low. None of the five assessors could achieve a correct "guess" rate of 90% or above. Only G.P.T. -2 Output Detector could achieve a higher correct "guess" rate of 90% for human exam writers. In addition, our results showed that there is no correlation between the guess made by the assessor and the actual writer of the question.

## Discussion

This is the first evidence in the literature showing that a commercially available, pre-trained A. I. can prepare exam material with a compatible quality with experienced human examiners.

MCQ is a crucial tool for assessment in education because they permit the direct measurement of many knowledge, skills, and competencies across a broad range of disciplines with the ability to test their concepts and principles, make judgments, drawing inferences, reasoning, interpretation of data, and information application [12, 14]. MCQs are also efficient to administer, easy to score objectively, and provide statistical information regarding the class performance on a particular question and assess if the question was appropriate to the context that was presented [21, 26]. A standard MCQ consists of the stem, the options, and occasionally auxiliary information [27]. The stem contains context, and content, and sets the question. The options include a set of alternative answers with one correct option and other incorrect options known as distractors [22]. Distractors are required to divert non-competent candidates away from the right answer, which serves as an essential hallmark of a high-quality question [19]. However, the major drawback to the MCQ format is that high-quality questions are difficult, time-consuming, and costly to write [28]. From our results, it is evident that even

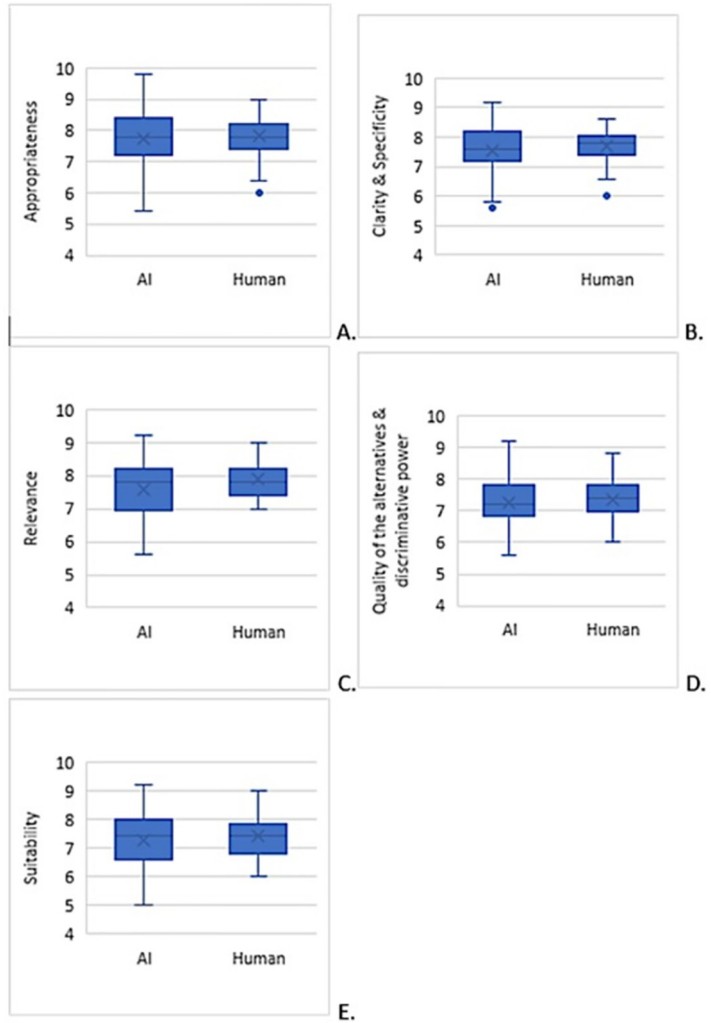

**Fig 3. Assessment scores across all five assessment domains.**

experienced examiners required more than ten minutes to prepare one question on average. With encouraging results demonstrated by the current study, we believe that A.I. could have the potential to generate quality MCQs for medical education. Artificial Intelligence (A.I.) has been used in education to improve learning and teaching outcomes for several years. The history of A.I. in education can be traced back to the 1950s when scientists and mathematicians

**Table 3. Comparison between AI vs human with the same reference.**

| | AI wins | Human wins | Equal | Mean difference (AI–human) (± SD) |
|---|---|---|---|---|
| Appropriateness of the question | 18 (36%) | 27 (54%) | 5 (10%) | - 0.11 ± 1.05 |
| Clarity and specificity | 18 (36%) | 26 (52%) | 6 (12%) | - 0.13 ± 1.08 |
| Relevance | 18 (36%) | 27 (54%) | 5 (10%) | - 0.32 ± 1.04 |
| Quality of the alternatives & discriminative power | 21 (42%) | 26 (52%) | 3 (6%) | - 0.10 ± 0.94 |
| Suitability for graduate medical school exam | 22 (44%) | 28 (56%) | 2 (4%) | - 0.14 ± 1.12 |
| Total score | 20 (40%) | 30 (60%) | 0 (0%) | - 0.80 ± 4.82 |

**Table 4. Blinded guess of question writer (i.e. AI vs human).**

|  | AI (total = 50) | Human (total = 50) | Correlation | p |
|---|---|---|---|---|
|  | (Correct guess, %) | (Correct guess, %) |  |  |
| Assessor A | 24, 48% | 23, 46% | - 0.14–0.26 | 0.55 |
| Assessor B | 14, 28% | 41, 82% | - 0.38–0.10 | 0.24 |
| Assessor C | 33, 66% | 24, 48% | - 0.35–0.06 | 0.16 |
| Assessor D | 27, 53% | 26, 52% | - 0.26–0.14 | 0.55 |
| Assessor E | 26, 52% | 32, 64% | - 0.36–0.04 | 0.11 |
| GPT-2 Output Detector | 7, 14% | 45, 90% | - 0.40–0.21 | 0.54 |

explored the mathematical possibility of A.I. [29]. In recent years, A.I. has been used to analyze student learning abilities and history, allowing teachers to create the best learning program for all students [5].

The introduction of ChatGPT, an AI-powered chatbot, has also transformed the landscape of A.I. in education. ChatGPT was trained on a large dataset of real human conversations and online data, leading to its capability of song or poem writing, storytelling, list creation, and even passing exams [10]. In our study, no additional training is required for the user or extra tuning of ChatGPT to yield similar results except for the need to follow the commands. As demonstrated in our study, a reasonable MCQ can be written by ChatGPT using simple commands with a text reference provided. The questions written by humans were rated superior to those written by A.I. when we compare the questions with the same reference head-to-head (Table 3). However, the difference in our raters' average scores was insignificant except in the relevance domain (Table 2). This shows that despite the apparent superiority of the MCQ written by humans, the score difference is actually narrow and mostly insignificant. This indicates a huge potential to explore the use of such tools as assistance in other educational scenarios.

However, ChatGPT and other AI-powered tools in education have also raised concerns about their negative impacts on student learning [11]. Including ChatGPT, most A.I. models are trained by the vast content available on the internet, and their reliability and credibility are questionable. Moreover, many A.I.s were found to have significant bias due to their training data [30]. Another major potential setback related to natural language generator A.I. is called hallucination [31]. Like hallucination described in humans, this condition refers to a phenomenon where the A.I. generates nonsensical, or unfaithful to the provided source input. This has led to immediate recall even for some initially promising A.I. from some of the largest internet companies, such as Galactica from Meta Inc [32]. Hence, our team proposed the use of the A.I. by educators, which demonstrate a feasible way of utilization for educational purpose. To minimize bias and hallucination, our proposed methodology consists of providing a reliable reference for the A.I. to generate questions instead of complete dependence on its database [33].

Compatible with the guidance from the Division of Education, American College of Surgeons, there were minimal negative features in both the MCQs written by ChatGPT and humans (14% (7/50) vs 12% (6/50)), and there was also no use of "except" or "All/none of the above", which could create additional confusion to exam candidates [20]. However, we also acknowledge that there were intrinsic limitations with ChatGPT in generating MCQs for medical graduate examinations. First, as it is a pure language-based model, it cannot create any text and correlate it with clinical photos or radiological images. This is also an essential area in the exam, assessing candidates' interpretation skills. In addition, our pre-study tests have found that ChatGPT performed poorly when it was instructed to generate a clinical scenario, possibly due to the high complexity of knowledge and experience required to create a relevant scenario,

limiting its use to assess candidates' ability of application of their knowledge. However, with the continuous effort from the OpenAI team and the rapid evolution of A.I. technology, these barriers might be solved in the near future [34].

## Limitations

The first limitation is that the reference material used was obtained directly from a textbook, and the length of the text was limited by the A.I. platform, which is currently at 510 tokens, potentially leading to selection bias by the operator. In contrast, human exam writers can recall information associated with the text, resulting in higher quality questions. Another limitation is the limited number of people involved and the number of questions generated, which could limit the applicability of the results. Only two professoriate staff from Medicine and Surgery departments, but not experts in all fields, developed the MCQs in this study. This differs from the real-life scenario where graduate exam questions were generated by a large pool of question writers from various clinical departments, which were then reviewed and vetted by a panel of professoriate staff. Besides, only 50 questions were generated by humans and another 50 by A.I., and only five assessors were involved. Real-world performance, in particular, the differentiate index, was impossible to assess due to the lack of actual students doing the test. Hence, efforts were made to improve the generalizability, including comprehensive coverage of all areas in both medicine and surgery and the participation of a multinational panel for assessment. Hence, efforts were made to improve the generalizability, including comprehensive coverage of all areas in both medicine and surgery and the participation of a multinational panel for assessment. The third limitation concerns the absence of human interference in the question-writing process. A.I. generated the questions, and the first-available question was captured without any polishing, while ChatGPT is also known to be sensitive to small changes in the prompt and can provide various answers even with the same prompt. And with extra fine-tuning in the form of further conversation with ChatGPT, the output quality from ChatGPT can be significantly enhanced. In addition, this study only evaluated the use of A.I. in generating MCQs. The full application of A.I. in developing the entire medical exam questions is yet to be thoroughly assessed. Lastly, with the improvements in the newer generation ChatGPT AI platform and other adjuncts, A.I. may perform better than what was observed in this current study. Nonetheless, this study provided solid evidence of the ability of ChatGPT and its strong potential in assisting medical exam MCQ preparation.

## Conclusion

This is the first study showing that ChatGPT, a widely available large language model, can be utilized as an exam question writer for graduate medical exams with comparable performance to experienced human examiners. Our study supports the continuous exploration of how large language model A.I. can assist academia in improving their efficiency while maintaining a consistently high standard. Further studies are required to explore additional applications and other limitations of the booming A.I. platforms to enhance reliability with minimal bias.

## Supporting information

**S1 Checklist. PLOS ONE clinical studies checklist.**
(DOCX)

**S1 File. ChatGPT webpage.**
(DOCX)

## Author Contributions

**Conceptualization:** Billy Ho Hung Cheung, Michael Tiong-Hong Co.

**Data curation:** Billy Ho Hung Cheung, Gary Kui Kai Lau, Gordon Tin Chun Wong, Elaine Yuen Phin Lee.

**Formal analysis:** Billy Ho Hung Cheung, Michael Tiong-Hong Co.

**Investigation:** Billy Ho Hung Cheung, Gary Kui Kai Lau, Gordon Tin Chun Wong, Elaine Yuen Phin Lee, Dhananjay Kulkarni, Choon Sheong Seow, Ruby Wong.

**Methodology:** Billy Ho Hung Cheung, Gary Kui Kai Lau, Choon Sheong Seow, Michael Tiong-Hong Co.

**Project administration:** Billy Ho Hung Cheung, Gary Kui Kai Lau, Gordon Tin Chun Wong, Elaine Yuen Phin Lee, Dhananjay Kulkarni, Ruby Wong, Michael Tiong-Hong Co.

**Resources:** Billy Ho Hung Cheung, Gordon Tin Chun Wong, Elaine Yuen Phin Lee, Dhananjay Kulkarni, Choon Sheong Seow.

**Software:** Billy Ho Hung Cheung.

**Supervision:** Gordon Tin Chun Wong, Michael Tiong-Hong Co.

**Validation:** Billy Ho Hung Cheung, Michael Tiong-Hong Co.

**Visualization:** Billy Ho Hung Cheung.

**Writing – original draft:** Billy Ho Hung Cheung.

**Writing – review & editing:** Billy Ho Hung Cheung, Michael Tiong-Hong Co.

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
