## [Decision Letter · Decision Letter 0]

17 Jul 2023

PONE-D-23-14620ChatGPT versus human in generating medical graduate exam questions – An international prospective studyPLOS ONE

Dear Dr. CO,

Thank you for submitting your manuscript to PLOS ONE. After careful consideration, we feel that it has merit but does not fully meet PLOS ONE’s publication criteria as it currently stands. Therefore, we invite you to submit a revised version of the manuscript that addresses the points raised during the review process.

We look forward to receiving your revised manuscript.

Kind regards,

Jie Wang, Ph.D.

Academic Editor

PLOS ONE

2. Thank you for including your ethics statement:  "N/A".  

a. For studies reporting research involving human participants, PLOS ONE requires authors to confirm that this specific study was reviewed and approved by an institutional review board (ethics committee) before the study began. Please provide the specific name of the ethics committee/IRB that approved your study, or explain why you did not seek approval in this case.

b. Please provide additional details regarding participant consent. In the ethics statement in the Methods and online submission information, please ensure that you have specified (1) whether consent was informed and (2) what type you obtained (for instance, written or verbal, and if verbal, how it was documented and witnessed). If your study included minors, state whether you obtained consent from parents or guardians. If the need for consent was waived by the ethics committee, please include this information.

“No”

“No”

6. We note that Figure 1 in your submission contain copyrighted images. All PLOS content is published under the Creative Commons Attribution License (CC BY 4.0), which means that the manuscript, images, and Supporting Information files will be freely available online, and any third party is permitted to access, download, copy, distribute, and use these materials in any way, even commercially, with proper attribution. For more information, see our copyright guidelines: http://journals.plos.org/plosone/s/licenses-and-copyright.

7. Please include your tables as part of your main manuscript and remove the individual files. Please note that supplementary tables (should remain/ be uploaded) as separate "supporting information" files

Additional Editor Comments:

The authors should pay careful attention to each of the comments below and address the issues raised by the reviewers. Since the scale of the current study is not sufficient to support a generic conclusion, the authors need to be more cautious when interpretting the results.

Reviewers' comments:

Reviewer's Responses to Questions

**Comments to the Author**

1. Is the manuscript technically sound, and do the data support the conclusions?

Reviewer #1: Yes

Reviewer #2: Yes

Reviewer #3: Yes

Reviewer #4: Partly

2. Has the statistical analysis been performed appropriately and rigorously? 

Reviewer #1: Yes

Reviewer #2: Yes

Reviewer #3: I Don't Know

Reviewer #4: No

3. Have the authors made all data underlying the findings in their manuscript fully available?

Reviewer #1: Yes

Reviewer #2: No

Reviewer #3: Yes

Reviewer #4: Yes

4. Is the manuscript presented in an intelligible fashion and written in standard English?

Reviewer #1: Yes

Reviewer #2: Yes

Reviewer #3: Yes

Reviewer #4: Yes

5. Review Comments to the Author

Reviewer #1: This is an interesting study. The actual quality of MCQ can be assessed by item analysis after testing it on students. The authors used assessment by experts. The study is interesting. needs some changes.

In title specify exam of which country? Different countries has different level of difficulty.

"An international prospective study" is not needed.

Background in abstract may have more detail about why this study was conducted.

Avoid old references.

Figure 3 is hazy.

Reviewer #2: My main criticism concerns the method used to assess the quality of the questions. For me the only scientifically valid way to evaluate the quality of the questions is to use them in real life, in official exams for example. Only the statistical analysis of students' answers makes it possible to evaluate the quality of a question. This is especially true for distractors. Making this criticism in the discussion is necessary in my opinion.

Reviewer #3: Many thanks for the authors to tackle this hot topic in medical education. I enjoyed reading your manuscript however, the following points should be addressed to improve it

1. The title “ChatGPT versus human in generating medical graduate exam questions” indicate more general term while the study mentioned only MCQ type of questions so either you need to include other types of questions or change the title to indicate the MCQ.

2. Method, how you make sure that the two faculty staff who generated the 50 questions were not using AI to generate questions? Clarify this in method section.

3. Some results mentioned in the discussion section “When the questions generated by ChatGPT were reviewed, we could also observe that they were compatible with the guidance from the Division of Education, American College ofSurgeons, with minimal negative features, including a minimal use of negative stem (only14% (7/50), compared to 12% (6/50) by human examiners) with a lack of “except”,“All/none of the above”

This should be mentioned in result section and discussed in the discussion section.

4. Regarding assessor evaluation of quality of questions it would be better to assess the Bloom levels of questions and add it as another parameter to ensure that the questions not only assessing the lower levels of Bloom (remembering and understanding) specially for questions generated by AI.

5. Table 3 need more clarification (how you calculate the percentage in each column)

6. Table 4 the calculation of % should be the number of truly guessed question written by AI to the 50 questions written by AI, the same for human correct guess(e.g for assessor B it should be 14/50 not 14/23

7. Attach a strobe checklist for the article and address the missed part in the manuscript.

8. Look for instruction of authors for both incitation and reference section.

Reviewer #4: The authors made a comparative study between ChatGPT and Human Experts in generating medical domain MCQ questions from textbooks. The study offers some insight on the performance of chatGPT in generating medical MCQs.

However, I don't find any technical contribution in this article. What is the contribution of the authors? They only made some comparison between the MCQs generated by ChapGPT and human experts using certain evaluation criteria.

Additionally, the article needs some improvement for common readers:

There is no explanation on how the questions are generated by ChatGPT. I feel a large number of readers do not have specific idea on how to generate MCQs from a textbook (as input) using ChatGPT. So, there should be some discussion on that.

What are the list of human input (like, specific information, selection of specific portion, any tuning, parameter setting) or amount of experience required to generate meaningful questions using ChatGPT? For instance, the authors mentioned, the length of the text was limited by ChatGPT at 510 tokens. What are the other such inputs?

The authors claimed that "There was no significant difference in question quality between questions drafted by AI versus human". This is actually the key highlight of the paper. However, this is not completely supported by evidence. When I study the values in Table 3, I find that the human experts are much better than ChatGPT. For instance, appropriateness of the question: 18 (AI) vs 27 (human); Clarity and specificity 18 (AI) vs 26 (human); Distractor quality 21 (AI) vs 26 (human). The gap is significant. Then how the claim is justified?

Also, the number of questions is too less to make such a generic comment on ChatGPT vs Human. The study should consider much larger number of questions, more subjects as input, more number of human experts.

More number of MCQ evaluation metrics are required to be considered for complete evaluation of the MCQs from all aspects. The evaluation of quality of MCQs is a tricky task, and various metrics have been used in the literature in the past two decades. For example, well-formed, sentence length, sentence simplicity, difficulty level, answerable, sufficient context, relevant to the domain, over-informative, under-informative, grammaticality, item analysis etc. are some metrics I find in the literature.

Finally, the human evaluators give a score against each evaluation metric in a scale of 1-10. What was the specific guidelines given to the evaluators for the scoring? Only the name of a evaluation metric might have certain ambiguity and might carry different meaning to different experts. So, there should be proper guidelines for the evaluators. Please mention those.

6. PLOS authors have the option to publish the peer review history of their article (what does this mean?). If published, this will include your full peer review and any attached files.

Reviewer #1: No

Reviewer #2: **Yes: **Pr. Olivier Palombi, Grenoble Université Alpes

Reviewer #3: **Yes: **Nazdar Ezzaddin Alkhateeb

Reviewer #4: No

---

## [Author Response · Author response to Decision Letter 0]

23 Jul 2023

Dear Editor-in-Chief and Reviewers,

Thank you very much for your kind consideration and comprehensive assessment of the article originally titled ChatGPT versus human in generating medical graduate exam questions – An international prospective study [PONE-D-23-14620] - [EMID:26d712de0fc6d895].

We have reviewed your valuable comments and recommendations and here is our reply and amendments.

1. Reviewer #1: This is an interesting study. The actual quality of MCQ can be assessed by item analysis after testing it on students. The authors used assessment by experts. The study is interesting. needs some changes.

In title specify exam of which country? Different countries has different level of difficulty.

"An international prospective study" is not needed.

Reply: 

TITLE

ChatGPT versus human in generating medical graduate exam multiple choice questions – A multinational prospective study (Hong Kong S.A.R, Singapore, Ireland, and the United Kingdom)

2. Background in abstract may have more detail about why this study was conducted.

Reply: 

ABSTRACT

Introduction

Large language models, in particular ChatGPT, have showcased remarkable language processing capabilities. Given the substantial workload of university medical staff, this study aims to assess the quality of multiple-choice questions (MCQs) produced by ChatGPT for use in graduate medical examinations, compared to questions written by university professoriate staffs based on standard medical textbooks.

3. Avoid old references

(Updated)

4. Figure 3 is hazy. 

We have revised the figure. Thank you.

5. Reviewer #2: My main criticism concerns the method used to assess the quality of the questions. For me the only scientifically valid way to evaluate the quality of the questions is to use them in real life, in official exams for example. Only the statistical analysis of students' answers makes it possible to evaluate the quality of a question. This is especially true for distractors. Making this criticism in the discussion is necessary in my opinion.

Reply: 

(Thank you for pointing this limitation out and our team is also aware of this limitation due to the lack of exam candidates for assessment. However, it might not be appropriate at this stage to apply these questions in real exams unless their usability has been assessed and further discussed in the faculty. Hence we think that it may be more suitable to address in the limitation session.)

Another limitation is the limited number of people involved and the number of questions generated, which could limit the applicability of the results…., and only five assessors were involved. Real-world performance, in particular, the differentiate index, was impossible to assessed due to the lack of actual students doing the test. Hence, efforts were made to improve the generalizability, including comprehensive coverage of all areas in both medicine and surgery and the participation of a multinational panel for assessment.

…

Reviewer #3: Many thanks for the authors to tackle this hot topic in medical education. I enjoyed reading your manuscript however, the following points should be addressed to improve it

6. (1.) The title “ChatGPT versus human in generating medical graduate exam questions” indicate more general term while the study mentioned only MCQ type of questions so either you need to include other types of questions or change the title to indicate the MCQ.

Reply: 

ChatGPT versus human in generating medical graduate exam multiple choice questions – A multinational prospective study (Hong Kong S.A.R, Singapore, Ireland, and the United Kingdom)

7. (2.) Method, how you make sure that the two faculty staff who generated the 50 questions were not using AI to generate questions? Clarify this in method section.

Reply:

Question construction 

…

Another 50 MCQs with four options were drafted by two experienced university professoriate staff (with more than 15 years of clinical practice and academic experience in Internal Medicine and Surgery, respectively) based on the same textbook reference materials.

They were given the exact instructions as the criteria mentioned above. Additionally, while they were allowed to refer back to additional textbooks or online resources, they were not allowed to use any AI tool to assist their question writing. 

…

8. (3.) Some results mentioned in the discussion section “When the questions generated by ChatGPT were reviewed, we could also observe that they were compatible with the guidance from the Division of Education, American College of Surgeons, with minimal negative features, including minimal use of negative stem (only14% (7/50), compared to 12% (6/50) by human examiners) with a lack of “except”,“All/none of the above”

This should be mentioned in result section and discussed in the discussion section.

Reply:

Results

….

In addition, when the questions generated by ChatGPT were reviewed, we observed few negative features, including a minimal use of negative stem (only 14% (7/50), compared to 12% (6/50) by human examiners) with a lack of “except”, “All/none of the above”.

Discussion

…

Compatible with the guidance from the Division of Education, American College of Surgeons, there were minimal negative features in both the MCQs written by ChatGPT and humans (14% (7/50) vs 12% (6/50)) and there was also no use of “except” or “All/none of the above”, which could create additional confusion to exam candidates 31.

9. (4.) Regarding assessor evaluation of quality of questions it would be better to assess the Bloom levels of questions and add it as another parameter to ensure that the questions not only assessing the lower levels of Bloom (remembering and understanding) specially for questions generated by AI.

Reply: 

Blinded assessment by multinational expert panel

and (V) Suitability for graduate medical school exam focused on the level of challenge including if the question higher order of learning outcomes such as application, analysis and evaluation, as elaborated by Bloom and his collaborators (24). 

…

10. (5. ) Table 3 need more clarification (how you calculate the percentage in each column)

Reply:

AI vs human

The average scores of the questions based on the same reading material were compared between that generated by AI vs by human writers. A "win" denotes a higher average score. Questions generated by human generally received a higher mark when compared to the AI counterpart (Table 3). However, AI-drafted questions outperformed human ones in 36% to 44% of cases across five assessment domains, including the total score.

11. (6.) Table 4 the calculation of % should be the number of truly guessed question written by AI to the 50 questions written by AI, the same for human correct guess(e.g for assessor B it should be 14/50 not 14/23

Reply:

Table 4. Blinded guess of question writer (i.e. AI vs Human)

 AI (total = 50)

(Correct guess, %) Human (total = 50)

(Correct guess, %) Correlation p

Assessor A 24, 48% 23, 46% - 0.14 – 0.26 0.55

Assessor B 14, 28% 41, 82% - 0.38 – 0.10 0.24

Assessor C 33, 66% 24, 48% - 0.35 – 0.06 0.16

Assessor D 27, 53% 26, 52% - 0.26 – 0.14 0.55

Assessor E 26, 52% 32, 64% - 0.36 – 0.04 0.11

GPT-2 Output Detector 7, 14% 45, 90% - 0.40 – 0.21 0.54

Reults

… The percentage of correct guesses was consistently low. None of the five assessors could achieve a correct “guess” rate of 90% or above. Only G.P.T. -2 Output Detector could achieve a higher correct “guess” rate of 90% for human exam writers….

12. (7.) Attach a strobe checklist for the article and address the missed part in the manuscript.

Reply:

Introduction

… We hypothesize that advanced large language models, such as ChatGPT, can reliably generate high quality MCQs that are comparable to those of an experienced exam writer….

Question construction

…ChatGPT was assessed in Hong Kong S.A.R. via a virtual personal network to overcome its geographical limitation (18)….

Blinded assessment by a multinational expert panel

…The question set were then sent to five independent international blinded assessors (From United Kingdom, Ireland, Singapore and Hong Kong S.A.R.) for assessment of the question quality (who were also the authors of this study as well)….

13. (8.) Look for instruction of authors for both incitation and reference section.

Reply: updated

Reviewer #4: The authors made a comparative study between ChatGPT and Human Experts in generating medical domain MCQ questions from textbooks. The study offers some insight on the performance of chatGPT in generating medical MCQs.

14. However, I don't find any technical contribution in this article. What is the contribution of the authors? They only made some comparison between the MCQs generated by ChapGPT and human experts using certain evaluation criteria.

Reply:

(The primary contribution of this study centers on the novel utilization of ChatGPT as a tool for multiple-choice question (MCQ) generation with the use of reference text as guidance (instead of posting a direct request to ask ChatGPT to write a question from scratch to improve its relevance and reliability), the importance of the request (also known as “prompt”, which people also developed a branch of speciality, coined “prompt engineering”), and its subsequent evaluation of its applicability. While I acknowledge the reviewer's perspective that the approach may not seem pioneering, given the widespread availability of ChatGPT, the application in this specific context is indeed unique. This work presents a valuable exploration of suitabilities in multiple dimensions, which will guide faculty members and academic staff in leveraging such technology to asssit their work. Moreover, this study also underscores the limitations inherent to this technology, offering a balanced understanding of its potential and constraints.)

Additionally, the article needs some improvement for common readers:

14. There is no explanation on how the questions are generated by ChatGPT. I feel a large number of readers do not have specific idea on how to generate MCQs from a textbook (as input) using ChatGPT. So, there should be some discussion on that.

Reply:

Question construction 

Fifty multiple choice questions (MCQs) with four options covering Internal Medicine and Surgery were generated by the ChatGPT with reference articles selected from the two reference medical textbooks. 

The commander using the ChatGPT will only copy and paste the instruction “Can you write a multiple-choice question based on the following criteria, with the reference I am providing for you and your medical knowledge? ” (also known as the prompt), the criteria and the reference text (selected from the reference textbook and input into ChatGPT interface as pure text input), to yield a question (Supplementary file 1). After sending the request to ChatGPT, a response will be automatically generated. Interaction is only allowed for clarification but not any modification. Questions provided by ChatGPT will be directly copied and used as the output for assessment by the independent quality assessment team. Moreover, a new chat session was started in ChatGPT for each entry to reduce memory retention bias.

….

Discussion

… 

As demonstrated in our study, a reasonable MCQ can be written by ChatGPT using simple command with a text reference provided. This indicates a huge potential to explore the use of such tool as an assistance in other educational scenarios.

….

15. What are the list of human input (like, specific information, selection of specific portion, any tuning, parameter setting) or amount of experience required to generate meaningful questions using ChatGPT? For instance, the authors mentioned, the length of the text was limited by ChatGPT at 510 tokens. What are the other such inputs?

Reply:

Discussion

…

In our study, no additional training is required for the user or extra tuning of ChatGPT to yield similar results except for the need to follow the commands. As demonstrated in our study, a reasonable MCQ can be written by ChatGPT using simple commands with a text reference provided. 

…

Limitations 

…

And with extra fine-tuning in the form of further conversation with ChatGPT, the output quality from ChatGPT can be significantly enhanced…

16. The authors claimed that "There was no significant difference in question quality between questions drafted by AI versus human". This is actually the key highlight of the paper. However, this is not completely supported by evidence. When I study the values in Table 3, I find that the human experts are much better than ChatGPT. For instance, appropriateness of the question: 18 (AI) vs 27 (human); Clarity and specificity 18 (AI) vs 26 (human); Distractor quality 21 (AI) vs 26 (human). The gap is significant. Then how the claim is justified?

Reply:

Discussion 

…

The questions written by humans were rated superior to those written by A.I. when we compare the questions with the same reference head-to-head (Table 3). However, the difference in our raters' average scores was insignificant except in the relevance domain (Table 2). This shows that despite the apparent superiority of the MCQ written by humans, the score difference is actually narrow and mostly insignificant. This indicates a huge potential to explore the use of such tools as assistance in other educational scenarios.

…

17. Also, the number of questions is too less to make such a generic comment on ChatGPT vs Human. The study should consider much larger number of questions, more subjects as input, more number of human experts.

(unfortunately the number of questions written and the assessors involved were limited by the resourced. We agreed that these limitations should be addressed in the Limitations session as well)

Limitations 

…

Another limitation is the limited number of people involved and the number of questions generated, which could limit the applicability of the results. Only two professoriate staff from Medicine and Surgery departments, but not experts in all fields, developed the MCQs in this study. This differs from the real-life scenario where graduate exam questions were generated by a large pool of question writers from various clinical departments, which were then reviewed and vetted by a panel of professoriate staff. Besides, only 50 questions were generated by humans and another 50 by AI, and only five assessors were involved. Efforts were made to improve the generalizability, including comprehensive coverage of all areas in both medicine and surgery and the participation of a multinational panel for assessment.

…

17. More number of MCQ evaluation metrics are required to be considered for complete evaluation of the MCQs from all aspects. The evaluation of quality of MCQs is a tricky task, and various metrics have been used in the literature in the past two decades. For example, well-formed, sentence length, sentence simplicity, difficulty level, answerable, sufficient context, relevant to the domain, over-informative, under-informative, grammaticality, item analysis etc. are some metrics I find in the literature.

Reply: 

(The domains for the assessment is specifically designed after literature review of relevant metrics for MCQ quality assessment and specific instructions were given to the assessors for the meaning of each domain.)

Blinded assessment by a multinational expert panel 

…

An assessment tool with five domains is specifically designed after literature review of relevant metrics for MCQ quality assessment in this study (20-23). Specific instructions were given the assessors for the meaning of each domain. These include (I) Appropriateness of the question, defined as if the question is correct, appropriately constructed with appropriate length and well-formed; (II) Clarity and specificity, defined as if the question is clear and specific without ambiguity, its answerability and without being under- or over-informative; (III) Relevance, defined as the relevance to clinical context; (IV) Quality of the alternatives & discriminative power for the assessment of the alternative choices provided; and (V) Suitability for graduate medical school exam focused on the level of challenge including if the question higher order of learning outcomes such as application, analysis and evaluation, as elaborated by Bloom and his collaborators (24). The quality of MCQs was objectively assessed by a numeric scale of 0 – 10. “0” is defined as extremely poor, while “10” means that it is at the quality of a gold-standard question. The total score of this section ranges from 0 to 50.

….

18. Finally, the human evaluators give a score against each evaluation metric in a scale of 1-10. What was the specific guidelines given to the evaluators for the scoring? Only the name of a evaluation metric might have certain ambiguity and might carry different meaning to different experts. So, there should be proper guidelines for the evaluators. Please mention those.

Reply:

Blinded assessment by a multinational expert panel 

… The quality of MCQs was objectively assessed by a numeric scale of 0 – 10. “0” is defined as extremely poor, while “10” means that it is at the quality of a gold-standard question. The total score of this section ranges from 0 to 50.

…

We hope that our reply address your concerns regarding our research and provide solid evidence of its originality, scientific robustness and importance for consideration for publication in your reputable journal. You are also very welcome to provide further comments and recommendations to improve the work in order to demonstrate its value towards potential audience.

Yours sincerely,

Dr Michael Co 

21/7/2023

---

## [Decision Letter · Decision Letter 1]

15 Aug 2023

ChatGPT versus human in generating medical graduate exam multiple choice questions – A multinational prospective study (Hong Kong SAR, Singapore, Ireland, and the United Kingdom)

PONE-D-23-14620R1

Dear Dr. CO,

We’re pleased to inform you that your manuscript has been judged scientifically suitable for publication and will be formally accepted for publication once it meets all outstanding technical requirements.

Kind regards,

Jie Wang, Ph.D.

Academic Editor

PLOS ONE

Additional Editor Comments (optional):

Reviewers' comments:

Reviewer's Responses to Questions

**Comments to the Author**

1. If the authors have adequately addressed your comments raised in a previous round of review and you feel that this manuscript is now acceptable for publication, you may indicate that here to bypass the “Comments to the Author” section, enter your conflict of interest statement in the “Confidential to Editor” section, and submit your "Accept" recommendation.

Reviewer #1: All comments have been addressed

Reviewer #2: All comments have been addressed

Reviewer #4: All comments have been addressed

2. Is the manuscript technically sound, and do the data support the conclusions?

Reviewer #1: Yes

Reviewer #2: Yes

Reviewer #4: Partly

3. Has the statistical analysis been performed appropriately and rigorously? 

Reviewer #1: Yes

Reviewer #2: Yes

Reviewer #4: No

4. Have the authors made all data underlying the findings in their manuscript fully available?

Reviewer #1: Yes

Reviewer #2: No

Reviewer #4: Yes

5. Is the manuscript presented in an intelligible fashion and written in standard English?

Reviewer #1: Yes

Reviewer #2: Yes

Reviewer #4: Yes

6. Review Comments to the Author

Reviewer #1: The study addresses an important and relevant topic concerning the potential use of AI language models for generating MCQs in medical education. The research purpose is well-defined, and the study's objectives are clear.

The study adopts a prospective design and employs a systematic approach by comparing ChatGPT-generated MCQs with those developed by human examiners. The use of standardized assessment scores and independent international assessors enhances the reliability and validity of the findings.

The article highlights the significant time advantage of ChatGPT in generating MCQs, with a total time of 20 minutes and 25 seconds to create 50 questions, compared to 211 minutes and 33 seconds taken by human examiners for the same task. This efficiency demonstrates the potential practicality of using AI in question generation.

The study reveals that ChatGPT-generated MCQs show comparable quality to those created by human examiners across most domains, including appropriateness, clarity, discriminative power, and suitability for medical graduate exams. This suggests that ChatGPT is capable of producing high-quality MCQs in these areas

Reviewer #2: (No Response)

Reviewer #4: Actually, I feel some more experiments are required to validate the claim. But it seems the authors feel their work is sufficient. Anyway ...

7. PLOS authors have the option to publish the peer review history of their article (what does this mean?). If published, this will include your full peer review and any attached files.

Reviewer #1: **Yes: **Dr. Himel Mondal

Reviewer #2: **Yes: **Full Professor Olivier Palombi, Université Grenoble Alpes

Reviewer #4: No

---

## [Editor Report · Acceptance letter]

18 Aug 2023

PONE-D-23-14620R1 

ChatGPT versus human in generating medical graduate exam multiple choice questions – A multinational prospective study (Hong Kong S.A.R., Singapore, Ireland, and the United Kingdom) 

Dear Dr. CO:

I'm pleased to inform you that your manuscript has been deemed suitable for publication in PLOS ONE. Congratulations! Your manuscript is now with our production department. 

Kind regards, 

on behalf of

Dr. Jie Wang 

Academic Editor

PLOS ONE